# FedDES: A Discrete-Event Simulator For Large-Scale Federated Learning

## Abstract

We introduce FedDES, a performance simulator for Federated Learning (FL) that leverages Discrete Event Simulation (DES) techniques to model key events—such as client updates, communication delays, and aggregation operations—as discrete occurrences in time. This approach accurately captures the runtime features of FL systems, providing a high-fidelity simulation environment that closely mirrors real-world deployments. FedDES incorporates all three known aggregation settings: Synchronous (e.g., FedAvg and FedProx), Asynchronous (e.g., FedAsync and FedFa), and Semi-Asynchronous (e.g., FedBuff and FedCompass). Designed to be framework-, dataset-, and model-agnostic, FedDES allows researchers and developers to explore various configurations without restrictions. Our evaluations involving over 1,000 clients with heterogeneous computation and communication characteristics demonstrate that FedDES accurately models event distribution and delivers performance estimates within 2% error of real-world measurements. While real-world workloads often take hours to evaluate, FedDES generates detailed, timestamped event logs in just few seconds. As a result, FedDES can significantly accelerate FL developing and debugging cycles, enabling developers to rapidly prototype and evaluate algorithms and system designs, bypassing the need for costly, time-consuming real-world deployments. It offers valuable performance insights—such as identifying bottlenecks, stragglers, fault-tolerance mechanisms, and edge-case scenarios—facilitating the optimization of FL systems for efficiency, scalability, and resilience.

## 1 Introduction

Federated Learning (FL) has emerged as a key paradigm in machine learning, primarily driven by growing concerns over privacy and the dwindling availability of publicly accessible datasets McMahan et al. (2017); Kairouz et al. (2021). Unlike traditional centralized learning approaches, FL allows multiple clients to collaboratively train a shared model while keeping their local data decentralized, addressing critical issues related to privacy and data sovereignty Yang et al. (2019). This decentralized learning approach is gaining widespread adoption across various domains, including healthcare, finance, and mobile applications, where data privacy and security are of paramount importance Bonawitz et al. (2019).

The field of FL is evolving rapidly, particularly in aggregation strategies. Aggregation strategies are being developed to tackle challenges such as data heterogeneity, fairness, privacy, and robustness, creating specialized aggregation techniques Li et al. (2020; 2021a). For instance, algorithms like `FedAvg` Karimireddy et al. (2020) and `FedProx` Li et al. (2020) have been introduced to address issues arising from non-IID data distributions and to improve robustness against stragglers and malicious clients. At the same time, scheduling algorithms are evolving toward semi-synchronous paradigms, which strike a balance between synchronous and asynchronous aggregations Li et al. (2023b); Chen et al. (2020). These semi-synchronous approaches optimize training speed and model accuracy while maintaining system stability. However, integrating such sophisticated scheduling mechanisms can fundamentally alter an FL framework's workflow, making it challenging to assess the effectiveness of new designs without comprehensive real-world evaluations.

In response to the rapidly changing landscape of FL, researchers are increasingly adopting fast-prototyping methodologies to iterate and refine their algorithms Li et al. (2019; 2020; 2021b).

Therefore, there is a growing recognition of the need for efficient and extensive simulation tools that can rigorously evaluate the robustness and efficacy of FL algorithms, particularly in areas such as fault tolerance and scalability in large-scale deployments Zhao et al. (2018). To meet this need, we propose **FedDES: A Discrete-Event Simulator for Federated Learning**, which applies Discrete Event Simulation (DES) principles to accurately model and analyze the intricate dynamics of FL systems Banks et al. (2010). FedDES captures the temporal evolution of FL processes by modeling client selection, local training, communication delays, and model aggregation as discrete events within a simulated timeline. This event-driven approach allows for precise temporal coordination and resource allocation, providing a detailed understanding of system behavior under various conditions. FedDES offers significant performance advantages inherent to DES, including high efficiency and scalability, enabling rapid simulation of FL workflows with thousands of clients exhibiting heterogeneous computational and communication capabilities. The simulator is framework-agnostic, allowing modeling and simulating distinct aggregation strategies across all known settings.

Moreover, FedDES is designed to be independent of specific datasets and models, allowing researchers to integrate their own data distributions and model architectures seamlessly without modifying the core simulation framework. We extensively evaluated FedDES on the NCSA Delta supercomputer, scaling real-world experimental events to over 1,000 clients with heterogeneous computational and communication settings. By comparing these results with our simulations, we demonstrated that FedDES achieves highly accurate event distribution and performance estimates, with an error margin of less than 2%. This underscores FedDES's ability to provide precise, large-scale event logs, facilitating debugging, bottleneck analysis, and rapid prototyping of new FL algorithms and system designs.

## 2 RELATED WORK

### 2.1 FL AGGREGATION STRATEGIES

Aggregation mechanisms critically influence the performance of FL systems. These mechanisms are typically classified into three categories:

**Synchronous Aggregation:** requires the server to wait for updates from all selected clients before aggregating and updating the global model. This ensures consistency but often leads to delays, especially when slow clients (stragglers) are involved. The most prominent synchronous algorithm, `FedAvg` McMahan et al. (2017); Karimireddy et al. (2020), aggregates client updates by computing a weighted average based on the number of samples each client holds. While effective for homogeneous training environments where each client's communication and communication resources are similar, `FedAvg` suffers from inefficiencies in environments with heterogeneous client resources, as the slowest client limits the overall training time.

**Asynchronous Aggregation:** update the global model without waiting for all clients, allowing faster updates but risking using stale client models. `FedAsync` Xie et al. (2019) addresses this by applying a staleness factor to penalize outdated client updates during aggregation. This method reduces waiting time, improving resource utilization. However, staleness penalties can cause the global model to drift away from the local data of slower clients, particularly in non-IID settings, affecting accuracy. `FedFa` Xu et al. (2024) takes asynchronous updates further by using immediate global updates without waiting for any client group. It mitigates the effects of stale updates with a sliding window aggregation technique, allowing the most recent updates to carry more weight. This approach boosts training speed significantly in heterogeneous environments but requires careful handling of staleness to maintain accuracy in non-IID settings.

**Semi-Asynchronous Aggregation:** combine aspects of both synchronous and asynchronous methods by grouping clients based on their computational capacity and scheduling updates within these groups. `FedBuff` Nguyen et al. (2022) introduces a buffering mechanism to temporarily store client updates and aggregate them once conditions are met, balancing update frequency and reducing the staleness of updates. However, managing the buffer requires heuristic tuning of the buffer size. On the other hand, `FedCompass` Li et al. (2023b) uses a computing power-aware scheduler to dynamically assign local steps to clients so that updates are received near synchrony within each group. This approach mitigates model staleness while maintaining efficiency, allowing `FedCompass` to

achieve faster convergence and higher accuracy in heterogeneous and non-IID settings than fully asynchronous methods.

## 2.2 FL SIMULATORS

FL simulators Beutel et al. (2020); Ryu et al. (2022); fls; flm; Li et al. (2023a); He et al. (2020); Sun et al. (2019); Ekaireb et al. (2022); Ro et al. (2021); Mugunthan et al. (2020) facilitate the validation of theoretical FL research. For example, Flower Beutel et al. (2020) integrates Apache Ray Moritz et al. (2018) to generate virtual clients, allowing for straightforward configuration of client resources like GPU memory ratios and CPU core allocations. Researchers can deploy custom aggregation strategies using selected models and datasets to evaluate statistical metrics such as convergence and accuracy. Furthermore, ns3-fl Ekaireb et al. (2022) combines FLsim with ns3, providing network configuration options that enable researchers to explore how network conditions affect FL performance. Our work complements existing FL simulators; FedDES is particularly effective for evaluating scheduling algorithms and delivering detailed performance metrics, including latency and straggler effects in dynamic environments.

## 2.3 DISCRETE EVENT-DRIVEN SIMULATION (DES)

DES models a system as a series of discrete events, each triggering state changes at specific times Banks (2005); Banks et al. (2010). DES has been widely applied in network simulations and distributed systems, where the timing and order of events are critical. Tools like ROSS Pearce (2002) and SimGrid Casanova (2001) demonstrate how DES can scale to millions of events and entities, providing insights into system performance in large-scale distributed environments. Our Fed-DES offers a significantly more scalable alternative by modeling FL workflows as discrete events. By simulating client-server activities and communication as state changes between events, FedDES scales efficiently to tens of thousands of clients while capturing key performance metrics, making it a robust tool for large-scale FL simulations.

## 3 PROPOSED FEDDES

### 3.1 COMMUNICATION-CENTRIC STATE MACHINE FOR FL EVENT MANAGEMENT

As illustrated in Figure 1, FedDES models the FL system as a state machine, where states are defined by computational workloads (e.g., local training, aggregation) and communication workloads (e.g., model parameter transmission). Client states track local models, computational capacity, and training progress, while the server maintains the global model and monitors client updates. Network states reflect link bandwidth, latency, and system topology, simulating real-world communication delays. State transitions are primarily driven by communication events (i.e., red actions in Figure 1, there are other transition events like the blue action, which depends on aggregation strategies), which act as synchronization points between clients and the server. For instance, after receiving the global model, a client transitions into the training state and, upon completion, triggers a communication event to send the updated model back to the server. Once sufficient updates are received upon criteria of different aggregation paradigms, the server transitions into the aggregation state. This communication-centric approach captures FL's asynchronous, distributed nature, where model exchanges rather than client computations govern state changes.

In FedDES, clients and servers transition between states such as idle, training, communicating, and aggregating, with communication events marking key interactions. This communication-driven design is well-suited for distributed asynchronous environments like FL for several reasons. First, it optimizes computational efficiency by focusing on client-server interactions, bypassing idle times. Second, communication is a natural trigger for state transitions, eliminating the need for global synchronization. Finally, it enables accurate simulation of network effects, such as bandwidth constraints and delays, which are critical to FL performance.

Event execution in FedDES is managed using a priority queue, advancing the simulation clock based on communication events. Key event types include *Client-Server Communication*, which simulates model transmission and accounts for network delays; *Aggregation*, where the server updates the global model; and *Client Computation*, which indirectly triggers communication events. By structuring the simulation around communication, FedDES efficiently captures the dynamics of large-

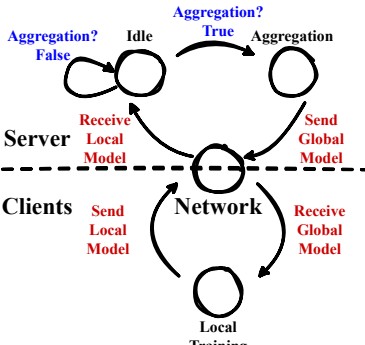

Figure 1: FedDES's State Machine Abstraction For FL

scale distributed FL systems, providing detailed performance metrics such as latency, throughput, and model convergence.

### 3.2 FEDDES FOR SYNCHRONOUS, ASYNCHRONOUS, AND SEMI-ASYNCHRONOUS FL

In FedDES, the server and the set of clients are implemented as discrete event-driven actors, where the server initiates the global model broadcast and manages client updates. Each client executes local training and communicates with the server. These actors are instantiated on separate virtual nodes to simulate distributed environments and communicate through a simulated network. The simulation is launched by creating the server actor on a central node and distributing the client actors across multiple nodes, reflecting real-world deployment. FedDES's simulation engine then handles the scheduling and execution of events for each actor.

Once the server and clients are launched, FedDES tracks each actor's state and manages their interactions using events. Each actor is responsible for processing its assigned tasks, such as communication, computation, and aggregation, based on the type of FL algorithm being simulated—synchronous, asynchronous, or semi-asynchronous. The following algorithms illustrate how these paradigms are modeled in FedDES.

**FedDES-Avg (Synchronous)**: `FedAvg` is a standard synchronous algorithm where the server waits for all clients to complete training before aggregation. In FedDES-Avg (Algorithm 1), the server distributes the global model (Event 1), clients perform local training (Event 2), and send updates back to the server (Event 3). Aggregation occurs once all updates are received (Event 4). Fed-DES introduces jitter and stragglers by introducing Gaussian (or can be any, depending on system characteristics) probability distribution for client computation and communication times, simulating real-world conditions.

**FedDES-Async (Asynchronous)**: In `FedAsync`, the server aggregates updates as soon as they arrive, without waiting for all clients. FedDES-Async (Algorithm 2) models this by having the server send the global model individually (Event 1). Clients perform training (Event 2), send updates (Event 3), and the server aggregates them on receipt (Event 4). Termination signals are sent when the simulation completes (Event 5). FedDES captures the impact of stale updates by modeling delays and jitters, revealing how asynchrony affects convergence and performance.

**FedDES-Compass (Semi-Asynchronous)**: `FedCompass` groups clients dynamically based on computational capacities, synchronizing updates within each group. FedDES-Compass (Algorithm 3) starts with the server broadcasting the global model to all clients (Event 1). Once a group completes training and sends updates (Event 3), the server aggregates the updates for that group (Event 4). Groups operate asynchronously relative to one another, balancing the efficiency of asynchronous updates with the consistency of synchronous aggregation. FedDES introduces varied delays across groups based on the compass scheduler's logic, simulating differences in computational speeds and network conditions. Although specific grouping and scheduling strategies may differ across semi-asynchronous algorithms, FedDES's flexible semi-async design abstracts the high-level grouping concept and can be easily adapted to other semi-asynchronous strategies.

---

**Algorithm 1:** FedDES-Avg (Exemplifies Synchronous FL)

---

**Input:** Global model $W_t$, Number of clients $n$, Number of rounds $T$, Communication cost $comm\_cost$, Training cost $train\_cost$, Aggregation cost $agg\_cost$

**Output:** Final global model after $T$ rounds

**1 Function** Server($W_t$, $n$, $T$, $dataloader\_cost$, $agg\_cost$, $comm\_cost$):
**2**     **for** $t = 1$ *to* $T$ **do**
**3**         **for** *client set $c_i$ from 0 to $n - 1$* **do**
**4**             $time+ = \text{CommTime}(W_t, c_i, comm\_cost)$    *(Simulate sending global model)*
**5**             **Event 1:** Distribute the global model to $c_i$
**6**         **end for**
**7**         **for** $c_i$ *from 0 to $n - 1$* **do**
**8**             $time+ = \text{CommTime}(W_{t+1}^{(i)}, \text{server}, comm\_cost)$    *(Simulate receiving local model)*
**9**             **Event 4:** Collect the local model from $c_i$
**10**         **end for**
**11**         $time+ = \text{AggregateTime}(n, agg\_cost)$    *(Simulate aggregation)*
**12**     **end for**
**13 Function** Client($c_i$, $n$, $T$, $dataloader\_cost$, $train\_cost$, $comm\_cost$):
**14**     **for** *each epoch $t = 1$ to $T$* **do**
**15**         $time+ = \text{CommTime}(W_t, \text{client}, comm\_cost)$    *(Simulate receive global model)*
**16**         **Event 2:** $c_i$ receives the global model from the server.
**17**         $time+ = \text{ComputeTime}(W_t, train\_cost)$    *(Simulate local training)*
**18**         $time+ = \text{CommTime}(W_{t+1}^{(i)}, \text{server}, comm\_cost)$    *(Simulate send local model)*
**19**         **Event 3:** $c_i$ sends the updated local model back to the server.
**20**     **end for**

---

**Algorithm 2:** FedDES-Async (Exemplifies Asynchronous FL)

---

**Input:** Global model $W_t$, Number of clients $n$, Communication cost $comm\_cost$, Training cost $train\_cost$, Aggregation cost $agg\_cost$

**Output:** Final global model after all updates

**1 Server**($W_t$, $n$, $agg\_cost$, $comm\_cost$)
**2 for** $c_i$ *from 0 to $n - 1$* **do**
**3**     $time+ = \text{CommTime}(W_t, c_i, comm\_cost)$    *(Send global model)*
**4**     **Event 1:** Send the global model to client $c_i$.
**5 end for**
**6 for** $t = 1$ *to* $n \times T$ **do**
**7**     **if** *local model is received from any client $c_i$* **then**
**8**         $time+ = \text{CommTime}(W_{t+1}^{(i)}, \text{server}, comm\_cost)$    *(Receive local model)*
**9**         **Event 4:** Receive and aggregate the local model from client $c_i$.
**10**         $time+ = \text{AggregateTime}(agg\_cost)$    *(Aggregation)*
**11**         **Event 1:** Send updated global model back to client $c_i$.
**12**         $time+ = \text{CommTime}(W_{t+1}^{(i)}, c_i, comm\_cost)$    *(Send updated model)*
**13**     **end if**
**14 end for**
**15 for** $c_i$ *from remaining unterminated clients* **do**
**16**     $time+ = \text{CommTime}(\text{Termination Signal}, c_i, comm\_cost)$
**17**     **Event 5:** Send termination signal to each client.
**18 end for**
**19 Client**($c_i$, $train\_cost$, $comm\_cost$)
**20 while** *termination signal not received* **do**
**21**     $time+ = \text{CommTime}(W_t, c_i, comm\_cost)$    *(Receive global model)*
**22**     **Event 2:** Client $c_i$ receives the global model from the server.
**23**     $time+ = \text{ComputeTime}(W_t, train\_cost)$    *(Local training)*
**24**     $time+ = \text{CommTime}(W_{t+1}^{(i)}, \text{server}, comm\_cost)$    *(Send local model)*
**25**     **Event 3:** Client $c_i$ sends the updated local model back to the server.
**26 end while**

---

---

**Algorithm 3:** FedDES-Compass (Exemplifies Semi-Asynchronous FL with Client Grouping)

---

**Input:** Global model $W_t$, Number of clients $n$, Number of epochs $T$, Communication cost $comm\_cost$,
      Training cost $train\_cost$, Aggregation cost $agg\_cost$, Maximum local steps $max\_local\_steps$,
      Group ratio $q\_ratio$, Group delay factor $\lambda$
**Output:** Final global model after all epochs

1   **Server($W_t$, $n$, $T$, $agg\_cost$, $comm\_cost$)**
2   $time+ = \text{CommTime}(W_t, \text{clients}, comm\_cost)$     *(Simulate communication)*
3   **Event 1:** Broadcast global model to all clients.
4   **while** $t < n \times T$ **do**
5      **if** *local model from client $c_i$ received* **then**
6          **Event 4:** Receive local model from client $c_i$.
7          Buffer client's model into its group.
8          **if** *group update is determined by compass algorithm or time limit reached* **then**
9             Perform group aggregation for the buffered models.
10            $time+ = \text{AggregateTime}(agg\_cost)$     *(Simulate aggregation)*
11            $t+ = |\text{group}|$
12            **Event 1:** Send updated global model back to all clients in the group.
13            $time+ = \text{CommTime}(W_{t+1}, \text{clients}, comm\_cost)$     *(Simulate communication)*
14          **end if**
15      **end if**
16   **end while**
17   **for** *$c_i$ from unterminated clients* **do**
18      $time+ = \text{CommTime}(\text{Termination Signal}, c_i, comm\_cost)$
19      **Event 5:** Terminate the remaining clients.
20   **end for**
21   **Client($c_i$, $n$, $train\_cost$, $comm\_cost$, $max\_local\_steps$)**
22   (Same as in Algorithm 2)

---

### 3.3 High-Usability: Real-World Simulation Methodology

As shown in Figure 2, FedDES enhances the usability and accuracy of FL simulations by integrating real-world considerations through a systematic process of logging, tracing, and profiling. The simulation framework incorporates real-world variability, such as system jitters, network delays, and stragglers, into its event generation process.

To achieve this, FedDES first logs and traces small-scale real-world FL workloads, capturing detailed event patterns such as local training, communication, and aggregation. These logs help abstract state machine transition rules for different FL paradigms. By understanding the timing and sequencing of these real-world events, FedDES accurately represents how the system transitions between states, including computation and communication phases.

FedDES also profiles computation and communication workloads. Profiling measures local training times, aggregation costs, and network delays for different system configurations. This data is then scaled to simulate large-scale deployments, with users only needing to input high-level parameters, such as the number of clients, system jitter distribution (like Gaussian), and network heterogeneity.

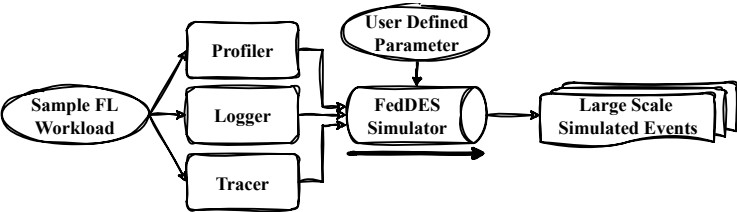

Figure 2: Compelete FedDES Architecture That Establishs An Systematic Simulation Workflow

This high-level input allows developers to easily generate accurate large-scale simulations without needing detailed knowledge of the system's internals. For example, users can simulate scenarios with varying system jitters or straggler behaviors by simply adjusting the parameters, while FedDES generates the corresponding detailed events and execution traces. Using profiling and tracing ensures

that FedDES produces accurate, real-world-like performance metrics, offering insights into how FL systems behave under different conditions.

Furthermore, all events in FedDES, including computation, communication, and aggregation, are logged with precise timestamps. This comprehensive event logging enables a deep analysis of system behavior, bottleneck identification, and performance evaluation across large-scale FL deployments. Integrating real-world profiling and tracing makes FedDES a highly usable and accurate tool for simulating and analyzing federated learning at scale.

## 4 CORRECTNESS OF FEDDES

This section proves FedDES's correctness for modeling and FL across three aggregation paradigms: synchronous, asynchronous, and semi-asynchronous. The correctness of FedDES is demonstrated by proving that the state transitions follow the definitions of each aggregation paradigm.

**Theorem 1.** FedDES-Avg correctly simulates synchronous FL by ensuring that the system maintains the invariant that no global model update occurs until all clients have completed their local training and sent their updates.

*Proof.* Section 3.1 formalizes FedDES's state machine for FL event management. The state machine is represented by the tuple:
$$\mathcal{M} = \langle \mathcal{S}, \mathcal{E}, \mathcal{T}, \mathcal{C}, \mathcal{A} \rangle$$
where $\mathcal{S}$ denotes system states (clients, server, network), $\mathcal{E}$ represents discrete events (communication, training, aggregation), $\mathcal{T} : \mathcal{S} \times \mathcal{E} \to \mathcal{S}$ is the state transition function, $\mathcal{C}$ are the conditions for state transitions, and $\mathcal{A}$ are actions triggered by events.

We define the state transitions for FedDES-Avg as:
$$\mathcal{T}_{\text{FedDES-Avg}} : S_{\text{server}}(t), S_{\text{client}_i}(t) \longrightarrow S_{\text{server}}(t+1), S_{\text{client}_i}(t+1),$$
where $S_{\text{server}}(t)$ is the server's state at time $t$, and $S_{\text{client}_i}(t)$ is the state of client $c_i$ at time $t$.

The correctness of FedDES-Avg is based on the following invariant:
$$\forall t, \quad C_{\text{all\_clients\_done}}(t) \implies \mathcal{A}_{\text{aggregate}}(W_{t+1}^{(1)}, \dots, W_{t+1}^{(n)}),$$
indicating that aggregation occurs only when all client updates are received.

1. Initial Condition: At $t = 0$, the server distributes the global model $W_0$ to all clients, triggering the state transition:
$$S_{\text{server}}(0) \to S_{\text{server}}(1), \quad S_{\text{client}_i}(0) \to S_{\text{client}_i}(1),$$
where $S_{\text{server}}(1)$ reflects the server waiting for client updates.

2. State Transition: For $t > 0$, each client $c_i$ completes local training and sends its updated model $W_{t+1}^{(i)}$ to the server. The condition $C_{\text{all\_clients\_done}}(t)$ is satisfied only when all client updates are received, triggering the aggregation action:
$$\mathcal{A}_{\text{aggregate}}(W_{t+1}^{(1)}, \dots, W_{t+1}^{(n)}).$$

3. Invariant Preservation: The transition function guarantees that the invariant is maintained at every time step. If any client $c_i$ has not completed its update, the server continues waiting, preventing premature aggregation and ensuring synchronous behavior.

Thus, FedDES-Avg correctly models synchronous FL by ensuring that no aggregation occurs until all client updates are received, preserving the synchronous nature of the system. $\square$

**Theorem 2.** FedDES-Async correctly simulates asynchronous FL by ensuring that each client update is aggregated as soon as it is received, without waiting for updates from other clients.

The proof of Theorem 2 is provided in Appendix A.1

**Theorem 3.** FedDES-Compass correctly simulates semi-asynchronous FL by ensuring that client updates within each group are synchronized while groups themselves operate asynchronously.

The proof of Theorem 3 is provided in Appendix A.2

## 5 EVALUATION

This section evaluates FedDES' accuracy by comparing the simulation's event distributions against real-world experimental results. We aim to determine how well the simulated FL events reflect the system dynamics compared to a real FL environment.

### 5.1 EXPERIMENTAL SETUP

**Testbed and Software Settings:** We use a Vision Transformer (ViT) Alexey (2020) on the partitioned *CIFAR-10* Krizhevsky et al. (2009) dataset for both real-world and simulated FL experiments. The real-world FL experiments involve more than 1,000 clients and three aggregation strategies: FedAvg, FedAsync, and FedCompass. Our FedDES simulations mirror these strategies using the *class* partitioning method. The real experiments are performed on the NCSA Delta High-Performance Computing (HPC) system, while the simulations run on a server equipped with two NVIDIA A40 GPUs and an Intel(R) Xeon(R) Gold 6336Y CPU. The real-world experiment captures communication events across multiple clients during an FL task, while the same FL task is simulated through FedDES with identical client configurations, communication steps, and task settings. In both the simulation and experiment, events are logged with timestamps, indicating when clients send or receive models and when the server performs aggregation.

**Metric Settings** We assess the similarity between the event distributions of real-world FL experiments and FedDES simulations across three dimensions: time, communication steps, and client ID. Evaluating the similarity between actual and simulated event distributions in only one or two dimensions (e.g., time alone or step-wise comparisons) would fail to capture the full complexity of the system. By analyzing the distribution of events across all three dimensions—time, step, and client ID—we ensure a more holistic and accurate comparison of the behavior between real-world experiments and simulations. The metrics we employ for this 3D evaluation are:

*Kullback-Leibler (KL) Divergence* Kullback & Leibler (1951): This metric measures how much the simulated distribution deviates from the real distribution across the three dimensions. A lower value indicates a higher similarity between the real and simulated data.

*Jensen-Shannon (JS) Divergence* Lin (1991): This symmetric measure compares the real and simulated distributions over time, step, and client ID. It captures the overall similarity of the distributions in three dimensions.

*Bhattacharyya Distance* Bhattacharyya (1943): The Bhattacharyya distance measures the overlap between the real and simulated distributions in the three-dimensional space of time, communication steps, and client ID. It is particularly sensitive to differences in distribution shape and spread.

By evaluating these metrics in three dimensions, we ensure that the simulation's accuracy is assessed not only with respect to time but also to the sequence of communication steps and the behavior of individual clients. This approach allows us to capture the full complexity of the system dynamics and ensures that the simulation faithfully reproduces the interaction patterns observed in real-world federated learning environments.

### 5.2 RESULTS AND ANALYSIS

Table 1, 2, and 3 compare the event distribution similarities between FedDES-Avg, FedDES-Async, and FedDES-Compass and their respective real workloads across three different simulation settings: simulation with no noise ($\mathcal{N}(0,0)$), simulation with Gaussian noise ($\mathcal{N}(0,0.12)$), and simulation with system heterogeneity observed from real workloads. We employ the discussed metrics to measure the divergence between the simulated and real workload distributions.

As shown in Table 1, across all three metrics (*KL, Jensen-Shannon, and Bhattacharyya*), the simulation with no noise ($\mathcal{N}(0,0)$) has the highest divergence from the real workload. Introducing Gaussian noise ($\mathcal{N}(0,0.12)$) slightly improves the similarity, as reflected by marginal decreases in all three metrics, suggesting that noise approximates the behavior of real systems better. The simulation with system heterogeneity further improves the similarity, showing the lowest values across all metrics. This demonstrates that accounting for system jitters and heterogeneity aligns the simulation more closely with real-world workloads. Figure 3 visualize the event distribution comparison among simulation settings (FedDES-Avg) and real-world experiments (FedAvg), showcasing visually per-

ceivable similarities and correct communication patterns in event distributions. When accounting for system jitter and heterogeneity, the simulation yields highly accurate execution time results in just **1.47s**, with a simulated execution time error of just **0.69%** (2336s simulated vs. 2320s real).

Table 1: Comparison of 3D event distribution evaluation in FedDES-Avg and real workloads across KL, Jensen-Shannon, and Bhattacharyya distances under different simulation settings.

| Simulation Settings / Metrics | Kullback-Leibler | Jensen-Shannon | Bhattacharyya |
|---|---|---|---|
| Simu. w/ $\mathcal{N}(0,0)$ | 9.125 | 0.495 | 1.25 |
| Simu. w/ $\mathcal{N}(0,0.12)$ | 9.121 | 0.494 | 1.249 |
| Simu. w/ system heterogeneity | 9.106 | 0.491 | 1.24 |

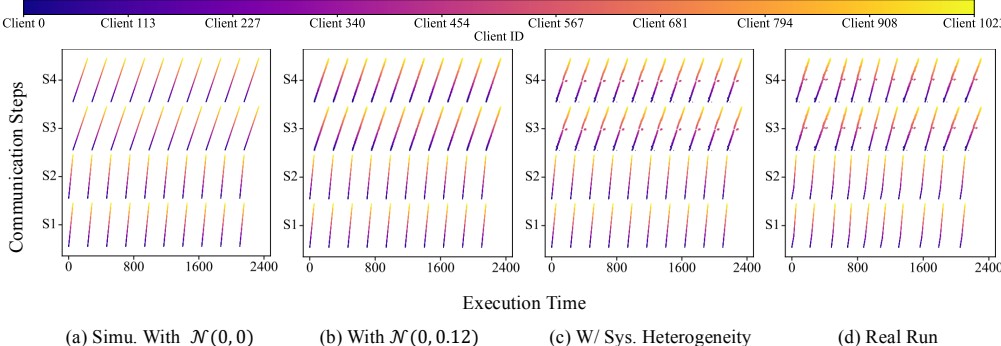

(a) Simu. With $\mathcal{N}(0,0)$      (b) With $\mathcal{N}(0,0.12)$      (c) W/ Sys. Heterogeneity      (d) Real Run

Figure 3: Event Distribution of FedDES for Synchronous FL training compared to Real Runs. The S1-4 in Communication Steps denote for: *S1*, Server sends global model; *S2*, Clients receive global model; *S3*, Clients send local model; *S4*, Server receives local model. Note that there's an explicit synchronization block after clients receive global model.

As shown in Table 2, similar to FedDES-Avg, the no-noise simulation shows the highest divergence (*KL:* 9.687, *JS:* 0.546, *Bhattacharyya:* 1.532). Introducing Gaussian noise improves the distribution similarity slightly, as indicated by minor reductions in *KL* and *Jensen-Shannon* Divergence, although *Bhattacharyya* Distance slightly increases. The simulation with system heterogeneity performs best, showing small reductions across all three metrics, indicating that heterogeneity captures real-world conditions better than noise alone. Figure 4 visualize the event distribution comparison among simulation settings (using FedDES-Async) and real-world experiments (FedAsync), which also showcased visually perceivable similarities and correct communication patterns in event distributions. When accounting for system jitter and heterogeneity, the simulation yields near-identical execution time results in just **2.03s**, with a simulated execution time error of just **0.04%** (2596s simulated vs. 2595s real).

Table 2: Comparison of 3D event distribution evaluation in FedDES-Async and real workloads across *KL*, *Jensen-Shannon*, and *Bhattacharyya* distances under different simulation settings.

| Simulation Settings / Metrics | Kullback-Leibler | Jensen-Shannon | Bhattacharyya |
|---|---|---|---|
| Simu. w/ $\mathcal{N}(0,0)$ | 9.687 | 0.546 | 1.532 |
| Simu. w/ $\mathcal{N}(0,0.12)$ | 9.721 | 0.547 | 1.544 |
| Simu. w/ system heterogeneity | 9.658 | 0.544 | 1.533 |

As shown in Table 3, for FedDES-Compass, we observe a larger difference in divergence. The system heterogeneity simulation shows the lowest divergence across all metrics, with *KL* divergence dropping to 8.709 from 9.591 in the no-noise setting. Gaussian noise also reduces divergence compared to the no-noise scenario. Still, the reduction is more significant in FedDES-Compass than in FedDES-Avg or FedDES-Async, indicating that this semi-asynchronous approach might be more sensitive to noise and system variations. The *Bhattacharyya* Distance improves significantly in the

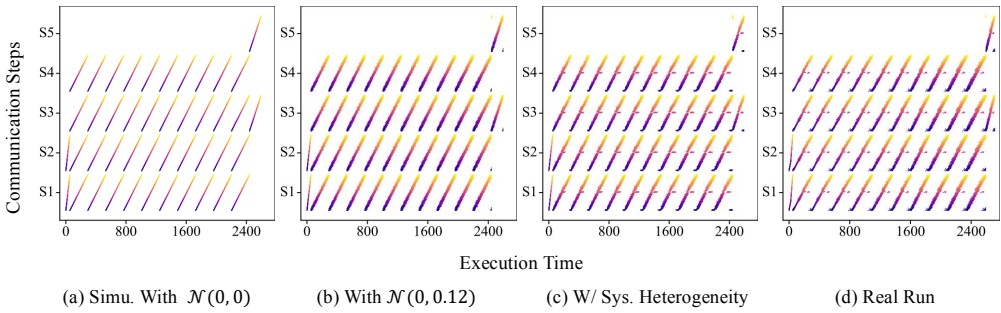

(a) Simu. With $\mathcal{N}(0,0)$      (b) With $\mathcal{N}(0,0.12)$      (c) W/ Sys. Heterogeneity      (d) Real Run

Figure 4: Event Distribution of FedDES for Asynchronous FL training compared to Real runs. Additionally, *S5* in Communication Steps denote for Server signals the finalization.

system heterogeneity setting, showing that this simulation closely approximates real-world event distributions. Figure 5 visualizes the event distribution comparison between simulation settings (using FedDES-Compass) and real-world experiments (FedCompass). Due to the dynamic grouping in FedCompass, which dynamically adjusts client training step size based on client speed, the event distribution exhibits higher randomness and sensitivity to system jitter and heterogeneity. Nonetheless, accounting for system jitter and heterogeneity in the simulation yields accurate results in only **2.77s**, with a simulated execution time error of just **1.95%** (2309s simulated vs. 2355s real).

Table 3: Comparison of 3D event distribution evaluation in FedDES-Compass and real workloads across *KL, Jensen-Shannon*, and *Bhattacharyya* distances under different simulation settings.

| Simulation Settings / Metrics | Kullback-Leibler | Jensen-Shannon | Bhattacharyya |
|---|---|---|---|
| Simu. w/ $\mathcal{N}(0,0)$ | 9.591 | 0.539 | 1.49 |
| Simu. w/ $\mathcal{N}(0,0.12)$ | 9.392 | 0.528 | 1.425 |
| Simu. w/ system heterogeneity | 8.709 | 0.489 | 1.214 |

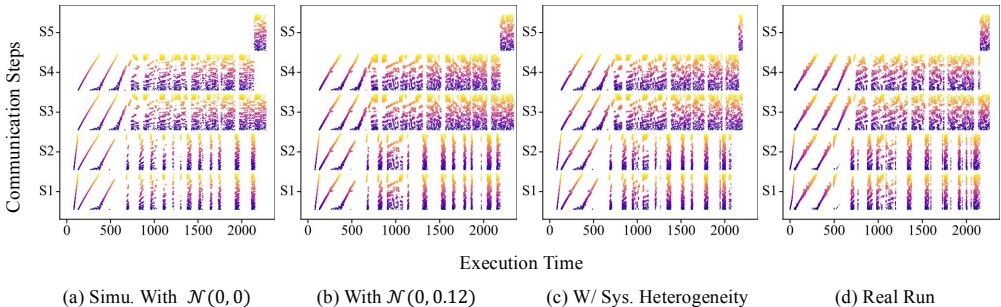

(a) Simu. With $\mathcal{N}(0,0)$      (b) With $\mathcal{N}(0,0.12)$      (c) W/ Sys. Heterogeneity      (d) Real Run

Figure 5: Event Distribution of FedDES for Semi-Async FL training compared to Real Runs. Note that clients have to explicitly block for synchronization before the amount of waiting clients reaches the preset threshold.

## 6 CONCLUSION

FedDES provides an efficient, scalable solution for simulating large-scale FL systems using Discrete Event Simulation (DES). By modeling client selection, training, communication, and aggregation as discrete events, it enables precise analysis of various FL strategies, including synchronous, asynchronous, and semi-asynchronous paradigms. FedDES is framework-agnostic, allowing researchers to integrate custom datasets and models, while accurately simulating heterogeneous client behaviors and network conditions. Evaluations on over 1,000 clients show FedDES delivers high accuracy, with less than 2% error compared to real-world experiments. This makes FedDES a powerful tool for debugging, performance analysis, and prototyping new FL algorithms.

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
