# A    APPENDIX

## A.1    CORRECTNESS PROOF OF THEOREM 2

*Proof.* The state transitions in FedDES-Async are defined as:

$$\mathcal{T}_{\text{FedDES-Async}} : S_{\text{server}}(t), S_{\text{client}_i}(t) \longrightarrow S_{\text{server}}(t+1), S_{\text{client}_i}(t+1),$$

where $S_{\text{server}}(t)$ represents the server's state at time $t$, and $S_{\text{client}_i}(t)$ represents the state of client $c_i$.

The correctness of FedDES-Async relies on the following invariant:

$$\forall t, \quad C_{\text{client\_ready}}(t) \implies \mathcal{A}_{\text{aggregate}}(W_{t+1}^{(i)}),$$

which states that the server aggregates the client $c_i$'s update immediately upon receiving it.

1. Initial Condition: At time $t = 0$, the server distributes the global model to clients, and each client begins local training. The server then enters a state of waiting for updates from any client.

2. State Transition: As soon as client $c_i$ completes its local training and sends its update to the server, the condition $C_{\text{client\_ready}}(t)$ is satisfied. The server aggregates the update immediately:

$$\mathcal{A}_{\text{aggregate}}(W_{t+1}^{(i)}).$$

3. Invariant Preservation: The transition function ensures that the invariant is maintained at each time step. Once a client $c_i$ sends its update, the aggregation occurs without waiting for updates from other clients. This behavior captures the asynchronous nature of the system.

Thus, FedDES-Async correctly models asynchronous FL by ensuring that updates are processed as soon as they are received. □

## A.2    CORRECTNESS PROOF OF THEOREM 3

*Proof.* The state transitions for FedDES-Compass are defined as:

$$\mathcal{T}_{\text{FedDES-Compass}} : S_{\text{server}}(t), S_{\text{client}_i}(t) \longrightarrow S_{\text{server}}(t+1), S_{\text{client}_i}(t+1),$$

where $S_{\text{server}}(t)$ is the server's state at time $t$, and $S_{\text{client}_i}(t)$ is the state of client $c_i$.

The correctness of FedDES-Compass relies on the following invariant:

$$\forall t, \quad C_{\text{group\_ready}}(t) \implies \mathcal{A}_{\text{group\_aggregate}}(\{W_{t+1}^{(i)} \mid i \in \text{group}\}),$$

which ensures that the server aggregates client updates within each group when the group is ready.

1. Initial Condition: The server broadcasts the global model to all clients at time $t = 0$. Clients are dynamically grouped based on their computational capacities, and each group operates semi-asynchronously with respect to others.

2. State Transition: When clients within a group complete their local training and send updates, the condition $C_{\text{group\_ready}}(t)$ is satisfied, triggering the group aggregation:

$$\mathcal{A}_{\text{group\_aggregate}}(\{W_{t+1}^{(i)} \mid i \in \text{group}\}).$$

3. Invariant Preservation: FedDES-Compass ensures that group updates are synchronized, meaning that the server aggregates the updates from all clients within a group before proceeding. However, groups themselves operate asynchronously, allowing for efficiency gains without sacrificing within-group consistency.

Thus, FedDES-Compass correctly models semi-asynchronous FL by maintaining synchronization within groups while allowing asynchrony across groups. □