# OpenReview forum: "FedDES: A Discrete-Event Simulator For Large-Scale Federated Learning"
_ICLR.cc/2025/Conference — ICLR 2025 Conference Withdrawn Submission_

### Official Review · Reviewer_nqA8 · 2024-11-01

**Soundness:** 2
**Presentation:** 3
**Contribution:** 2
**Rating:** 5
**Confidence:** 4

**Summary:**

This paper introduces FedDES, a discrete-event simulator designed to model and evaluate federated learning systems at scale. The key innovation is using discrete event simulation (DES) techniques to model FL events like client updates, communication delays, and aggregation operations. FedDES supports all three major FL aggregation paradigms: synchronous, asynchronous, and semi-asynchronous. The authors demonstrate that FedDES can accurately simulate large-scale FL deployments with over 1,000 clients while producing results within 2% error of real-world measurements.

**Strengths:**

1.Technical Innovation: Novel application of DES principles to FL simulation, providing a systematic way to model complex FL system dynamics；Comprehensive support for all three FL aggregation paradigms
in a single framework
2.Theoretical Foundation:Rigorous mathematical proofs of correctness for all three aggregation paradigms；Well-defined state machine abstraction for FL event management；Clear formalization of system states, events, and transitions
3.Implementation & Evaluation:Thorough experimental validation using real-world FL workloads；Impressive scale testing with over 1,000 clients；Detailed analysis using multiple metrics across three dimensions Strong accuracy with <2% error compared to real-world results

**Weaknesses:**

1.Limited Discussion of Scalability Bounds:The paper doesn't thoroughly explore the upper limits of FedDES's scalability；No clear discussion of memory requirements or computational complexity.
2.Validation Scope:Evaluation focuses mainly on Vision Transformer with CIFAR-10;Could benefit from testing with more diverse models and datasets; Limited exploration of edge cases or failure scenarios
3.Network Modeling:While network delays are considered, there's limited discussion of complex network topologies or dynamic network conditions; The paper could benefit from more detailed analysis of how network conditions affect simulation accuracy

**Questions:**

1.Why was Vision Transformer on CIFAR-10 chosen as the primary test case? Usually SOTA methods use more complex dataset. I wonder if other models/datasets could be tested? If so, what will be the results?

2.How does FedDES handle client failures during simulation? What recovery mechanisms are implemented?

3.Could you provide more details about the methodology for transferring real-world heterogeneity patterns to the simulator?

4.Are there plans to support more advanced FL techniques (personalization, compression)?

---

### Official Review · Reviewer_3tth · 2024-11-03

**Soundness:** 1
**Presentation:** 2
**Contribution:** 1
**Rating:** 3
**Confidence:** 4

**Summary:**

This paper proposes a discrete event simulator for FL. Experiments are conducted to evaluate the accuracy of the simulator compared with real world experiments.

**Strengths:**

Implemented a discrete event simulator and made efforts to achieve good simulation accuracy.

**Weaknesses:**

There are several significant weaknesses in the current version of the paper.

The paper claims that FedFS can help accelerate FL developing and debugging cycles. The key advantage should be providing modular design and clear interfaces, allowing the FL developers to interact with the simulator and to plug in their own FL protocols, optimization strategies and datasets. However, this paper does not present any of these aspects.

In section 4, it states that Section 3.1 formalizes FedDES’s state machine for FL event management. But this is presented in a descriptive manner rather than formally. As a result, the subsequent theorems and proofs lack solid theoretical foundation and depth.

I also have the reservation whether this work, developing a FL simulator, falls into the scope of ICLR.

**Questions:**

Does FedDES provide flexibility and interfaces for the users to interact with it?

Why are both KL and JS divergence used to evaluate the simulator?

Has FedDES been open-sourced?

---

### Official Review · Reviewer_kPRj · 2024-11-08

**Soundness:** 3
**Presentation:** 3
**Contribution:** 1
**Rating:** 3
**Confidence:** 3

**Summary:**

This paper designs a new federated learning (FL) simulator named FedDES, where it uses discrete event simulation techniques to model key events in FL training, such as client local updates, communication delays and aggregation operations. FedDES includes three algorithms to capture the FL applications in different setups. The performance of FedDES has been evaluated by a large computing cluster.

**Strengths:**

* The paper is, in general, easy to follow.
* The proposed training pipeline is intuitive.

**Weaknesses:**

* My major concern is the contribution of the paper. At a high level, this is a simulator with the aim of generating almost accurate results after real-world deployment. Yet, it remains unclear what the real-world deployment environment means.
* Next, the authors claim that FedDES generates event logs in a few seconds. However, the contribution here seems to be incremental. It is a must for every simulator, not only FL, to log data in detail for faster debugging.
* The related work seems to be a bit off either. Section 2.1, which is tiltled FL aggregation strategies, describes many methods that differ from each other far beyond aggregation themselves. In addition, although the FL algorithms' categories appear intuitive, the authors fail to capture the key components that differentiate different categories and affect their implementations of the FL simulator. Moreover, given the rapid development of FL research, the citations inside are far from enough (now is ~2 per strategy).
* In conventional FL setups and in most of the machine learning tasks, the trainers have to follow the protocols set by the algorithms. I failed to find something novel unless I missed something.
* The pseudocode in Algorithm 1-3 is not helpful. This is because most of the lines capture how the clients/ server time the training progresses without contextualizing what the real problem is there.
* The theorems proved in Section 4 seem to be trivial since all the simulators need to follow the rules as specified by the algorithms.
* In Section 5, the implications of three divergence measures are almost the same. The authors need to provide performance comparisons with the other FL simulators in terms of the 3D things. The current evaluations seem to be narrow.
* In Section 5, the definitions of real-world deployment are still missing. It also remains unclear whether FedDES can simulate a given type of GPU simulation results or whether it is suitable for a family of hardware devices.
* In Section 5, beyond purely comparing the fast implementation speed of FedDES, the authors also need to verify its correctness by testing the same client population, neural network models, and local datasets across different simulators.

**Questions:**

* Can the authors elaborate on the definitions of the real-world deployment?
* In Section 2.2, the authors say "our work complements existing FL simulators...delivering detailed performance metric." Can the authors elaborate on that by making a more detailed comparison table with prior literatures?
* In Section 3.3, the authors argue that FedDES captures the logs during small-scale FL workloads. While I agree that they can be referenced during large-scale training, how do the authors deal with the scaling client population? Is there an exact formula where we can translate the small-scale training logs into large-scale training scenarios?
* In Section 3.3, what is the definition of system jitters?
* How do the authors construct the non-IIDness across different clients?
* Can the authors provide comparison results with other FL simulators, such as flower? Without context, it is hard to verify the efficiency claim as stated in the paper.

---

### Note · Authors · 2024-11-18

I have read and agree with the venue's withdrawal policy on behalf of myself and my co-authors.